# Ultra-fast green hydrogen production from municipal wastewater by an integrated forward osmosis-alkaline water electrolysis system

Gabriela Scheibel Cassol [1], Chii Shang [1,2], Alicia Kyoungjin An [3], Noman Khalid Khanzada[3,4], Francesco Ciucci [5,6], Alessandro Manzotti[5], Paul Westerhoff [7], Yinghao Song [1] ✉ & Li Ling [8] ✉

Recent advancements in membrane-assisted seawater electrolysis powered by renewable energy offer a sustainable path to green hydrogen production. However, its large-scale implementation faces challenges due to slow power-to-hydrogen (P2H) conversion rates. Here we report a modular forward osmosis-water splitting (FOWS) system that integrates a thin-film composite FO membrane for water extraction with alkaline water electrolysis (AWE), denoted as FOWS$_{AWE}$. This system generates high-purity hydrogen directly from wastewater at a rate of 448 Nm$^3$ day$^{-1}$ m$^{-2}$ of membrane area, over 14 times faster than the state-of-the-art practice, with specific energy consumption as low as 3.96 kWh Nm$^{-3}$. The rapid hydrogen production rate results from the utilisation of 1 M potassium hydroxide as a draw solution to extract water from wastewater, and as the electrolyte of AWE to split water and produce hydrogen. The current system enables this through the use of a potassium hydroxide-tolerant and hydrophilic FO membrane. The established water-hydrogen balance model can be applied to design modular FO and AWE units to meet demands at various scales, from households to cities, and from different water sources. The FOWS$_{AWE}$ system is a sustainable and an economical approach for producing hydrogen at a record-high rate directly from wastewater, marking a significant leap in P2H practice.

The utilisation of renewable energy in the process of water electrolysis to produce green hydrogen offers significant promise for decarbonisation in many sectors. It is projected that by the year 2050, around 86% of the entire global annual power generation, amounting to 55,000 TWh, will be derived from renewable energy sources[1]. However, it is worth noting that up to 40% of the overall renewable electricity may be wasted due to the intermittent and variable characteristics inherent in renewable energy production[2–4]. The power-to-hydrogen (P2H) system provides a flexible approach to use renewable energy to produce green hydrogen (H$_2$) via water electrolysis to minimise renewable energy wastage. Currently, over 400 P2H projects are being planned or constructed using renewable energy, mainly in European countries (comprising ~70%) (Fig. 1a), for completion by 2030. P2H projects are also expected by 2050 in North African countries, India, China, Chile, Australia, and Saudi Arabia, driven by their abundant renewable resources and diverse strategies towards a low-carbon economy[1,5,6]. However, to limit global warming to well below 2 °C and achieve climate neutrality by 2050[7], electrolyser capacity

**Fig. 1 | Water risk atlas with planned H₂ projects and the FOWS_AWE system.**
**a** The distribution of hydrogen projects worldwide is categorised in small-scale (<100 MW), medium-scale (100–1000 MW), and large-scale (>1000 MW) according to water risk. The map was built using Geographic Information System (GIS) software by combining data from two global databases, i.e. water-related risk information across large geographies from the World Resources Institute (WRI) Aqueduct[55] and a worldwide database of hydrogen projects compiled by the International Energy Agency (IEA)[56] **b** Schematic diagram illustrating the FOWS_AWE system for green hydrogen production from municipal wastewater effluent.

must increase 6000–8000-fold[7,8]. Such high green H₂ yields to meet city-scale energy require large amounts of water (i.e. 10–15 L of water per kg of H₂). Water is particularly valuable and crucial in areas prone to the risks of water scarcity or quality deterioration, including Northern China, the Middle East, North Africa, and India (Fig. 1a)[9]. However, it is noteworthy that around 30% of ongoing or planned P2H projects are located in regions where populations facing water scarcity are projected to increase up to 154% by 2050[10]. These issues pose a potential constraint on the wide application of the P2H strategy[11].

Several strategies propose alternative water sources, mostly seawater, for utilisation in P2H systems. Direct seawater electrolysis results in limited electrolyser lifespan and low H₂ production rates due to various factors including the precipitation of cations such as calcium and magnesium at the cathode, electrode corrosion, and the formation of parasitic chlorine by-products at the anode[12–15]. Seawater desalination via reverse osmosis (RO) before electrolysis is often employed for H₂ production, but it is hampered by membrane fouling[16,17]. Incorporating forward osmosis (FO) before RO (FO–RO) mitigates membrane fouling, but it increases costs due to the secondary processes required to separate the clean water and regenerate the draw solution[17–19].

Recent advancements in membrane-assisted seawater electrolysis offer a promising future for P2H solutions. By incorporating a hydrophobic polytetrafluoroethylene (PTFE) membrane, water can be extracted from seawater as vapour which subsequently undergoes hydrolysis[20]. Similarly, coupling FO to water electrolysis (FOWS) can extract water under an osmotic gradient, and the extracted water is sequentially hydrolysed to sustain continuous operation[21]. Such innovative systems eliminate the need for energy-intensive RO and the additional steps to separate clean water before entering the electrolyser, thus reducing capital and operational costs (Supplementary Table 2)[17,21,22]. However, both systems yield H₂ at <30 Nm³ day⁻¹ m⁻² of membrane area, due to the low vapour pressure gradient driving the water flux across the PTFE membranes, the low conductivity of pH-neutral draw solutions in FO for electrolysis, and the high osmolarity of seawater, consequently restricting fast P2H conversion and its large-scale application[21–23]. We hypothesise that utilising potassium hydroxide (KOH) for FOWS is more efficient because KOH ensures high current densities to split the water via alkaline water electrolysis (AWE) and may serve as a suitable draw solution for FO to provide an osmotic gradient to extract water if coupled with a KOH-compatible FO membrane. Although acidic media may also enhance FOWS, using KOH is preferred as it allows the use of cost-effective materials (i.e. nickel) and minimises chlorine by-products due to higher overpotential and slower chlorine evolution reaction (CER) kinetics[24,25]. Further, the technological maturity of AWE compared to the acidified ones ensures superior safety, reliability, and scalability. We also hypothesise that wastewater effluent as an alternative water source is better than

seawater, as it enhances water production due to its lower osmolarity, presents less chloride for CER, and generates less hazardous brine which is costly for disposal[26–29]. Additionally, using wastewater effluent is more practical than seawater, especially for inland communities where water resources are scarce and seawater is unavailable but where wind and solar farms can provide green electricity to drive P2H systems[30]. This background information addresses the need to better understand the selection of FO membrane materials and electrolytes, and the impacts of impurities in wastewater effluent on the $H_2$ production rate of the FOWS system, and the operational versatility and stability of the system.

Here, we describe a FOWS$_{AWE}$ system that integrates FO and AWE utilising KOH as the shared operational solute to enable continuous $H_2$ production directly from wastewater effluent (Fig. 1b), achieving an impressive $H_2$ production rate of 448 Nm$^3$ day$^{-1}$ m$^{-2}$ of membrane area, over 14 times faster than the state-of-the-art seawater electrolysis[21,22]. Such fast $H_2$ production arises from the employed semi-permeable thin-film composite (TFC)-FO membrane, which tolerates KOH concentrations up to 1 M, allowing the use of KOH as a draw solution to enhance water production from wastewater and as the optimal electrolyte to produce $H_2$. The FOWS$_{AWE}$ system also shows consistent stability, maintaining a stable cell voltage of $1.79 \pm 0.01$ V during the entire testing period and a high Faradaic efficiency of ~99% even with some impurities from wastewater permeating through the FO membrane. It achieved a specific energy consumption (SEC) (i.e. average energy consumption for producing Nm$^3$ of $H_2$) of ~4.43 kWh Nm$^{-3}$ at 23 °C, which is comparable with commercial alkaline electrolysis using deionised water, and further reduced to 3.96 kWh Nm$^{-3}$ at 40 °C. Moreover, the highly modular FO and AWE units, in combination with our developed water-hydrogen balance model, allow for equalising the rates of water production from FO and those of water consumption from AWE. This flexibility allows the FOWS$_{AWE}$ system design to meet various scales with different water sources. Thus, our approach for fast and energy-efficient $H_2$ production from wastewater by the highly modularised FOWS$_{AWE}$ system demonstrates a promising strategy for the large-scale and sustainable P2H practice.

## Results and discussion

### Producing clean water via FO using KOH as a draw solution

To integrate FO with AWE, KOH and two other electrolytes including sodium bicarbonate (NaHCO$_3$), and potassium pyrophosphate (K$_4$P$_2$O$_7$) were considered as potential draw solutions for FO instead of the conventional NaCl, as NaCl induces a CER, which negatively affects $H_2$ generation and damages both the FO membrane and electrodes[30–33]. Among these electrolytes at a concentration of 1 M, KOH shows current densities 18 and 12 times higher than those of NaHCO$_3$ and K$_4$P$_2$O$_7$ at 2 V, respectively (Supplementary Fig. 1). The higher current density of KOH represents a higher $H_2$ production rate in AWE compared to the other two electrolytes. To not compromise the AWE performance, we tested if KOH could be used as the draw solution in FO for extracting water from simulated wastewater effluent containing sodium acetate (NaAc), ammonia/ammonium ions (NH$_3$/NH$_4^+$), and chloride. Besides, although the current density of KOH increases with increasing concentration[34], KOH at concentrations higher than 1 M causes the pH of the draw solution to exceed 14, which carries the risk of damaging commercial TFC-FO membranes. Therefore, we tested 1 M KOH to evaluate its impact on the TFC-FO membranes (details given in the *Methods* section).

Figure 2a shows the water flux (i.e. the permeated water volume per unit of time and per unit of membrane areas, L h$^{-1}$ m$^{-2}$ (LHM)) and reverse salt flux (RSF) (i.e. draw solute permeated to the feed solution side per unit of time and per unit of membrane areas, g h$^{-1}$ m$^{-2}$ (GHM)) of the FO process using 1 M KOH as the draw solution for five cycles of five hours each. The water flux decreased from $17.2 \pm 0.3$ to $14.9 \pm 0.1$ LHM after 5 h in each cycle, while the RSF remained constant

at $-121 \pm 3$ GHM. The water flux is comparable to a system using NaCl as a draw solution (Supplementary Fig. 2), indicating that the KOH is an effective draw solution to extract water through FO processes. The primary concern associated with the use of KOH as a draw solution is its potential to damage the FO membrane, as the reverse flux of KOH may cause hydrolysis of the polymer chains that constitute the membrane selective layer (on the feed side)[35]. The integrity of the commercial TFC-FO membrane after operating in 1 M KOH solution was examined by comparing the water flux and RSF of the used membrane after the tests shown in Fig. 2a and a new pristine membrane, both using deionised water as the feed solution and 1 M NaCl as the draw solution (Fig. 2b). Figure 2b shows that the water flux of the used membrane was reduced by less than 5% compared to the pristine membrane, while the RSF of the two membranes was comparable. The membrane integrity in terms of water flux and RSF was not compromised. In addition, results from scanning electron microscopy (SEM) revealed no change in the "leaf-like" surface of the selective layer on the unused membrane and the used membrane after five cycles (Fig. 2c, d and Supplementary Fig. 3a, b) and in the peaks of the functional groups in the Fourier transform infra-red spectroscopy (FTIR) (Supplementary Fig. 4). The comparable water flux and RSF between the pristine and the used membrane suggest that the FO membrane was not damaged by KOH. The high RSF may be attributed to the smaller hydrated ionic size and higher ionic mobility of KOH[36], or the ionisation of the membrane via deprotonation of the functional groups on its surfaces facilitating the transport of cations across the membrane[37,38]. Nonetheless, the RSF accounted for less than 1.5% of the initial KOH concentration after a cycle, suggesting the impact due to this issue was minor. Moreover, since KOH is inexpensive, any losses can be easily replenished in the system.

Since impurities in the wastewater effluent may permeate through the FO membrane and thus affect electrolysis, we examined the rejection of common impurities in the wastewater effluent by the FO membrane, including NaAc, NH$_3$/NH$_4^+$, and chloride (Fig. 2e). The rejection rates of NaAc, NH$_3$/NH$_4^+$, and chloride remained stable throughout the five cycles at about 85%, 63%, and 98%, respectively. The lower rejection of NH$_3$/NH$_4^+$ occurred as the TFC-FO membrane surface is negatively charged, which allows anions to be intercepted more readily than cations or unionised NH$_3$ which is the dominant form of NH$_3$/NH$_4^+$ at high pH. The rejection of additional common impurities found in wastewater and varied pH levels was also examined, with rejection rates exceeding 90% (Supplementary Figs. 5 and 6). The potential impacts to the electrolysis process from the permeated impurities were then investigated using a commercial 5-cell alkaline stack with nickel alloy-based electrodes by comparing the current density–voltage (J–V) curves generated from: (i) 1 M KOH, (ii–iv) 1 M KOH mixed separately with NaAc, NH$_3$/NH$_4^+$, and chloride at a concentration of 10 mM for studying their accumulation effect, respectively, or (v) a 1 M KOH mixture consisting of 1.1 mM of NaAc, 0.7 mM NH$_3$/NH$_4^+$ as N, and 0.08 mM chloride observed in the permeate from the rejection test, as shown in Fig. 2f. All tested solutions exhibited a consistent behaviour in their J–V curve performances, suggesting that the presence of NaAc, NH$_3$/NH$_4^+$, and chloride in KOH minimally impacted the current densities. We infer that the high concentration of KOH, sustaining a high pH (>8), mitigated the negative effects of NaAc and NH$_3$/NH$_4^+$ and suppressed side reactions such as chlorine evolution[31,39]. Therefore, the above results of the high and constant water flux, the intact FO membrane, and the minor impact of impurities permeated from the FO membranes on electrolysis suggested using KOH as the draw solution in the FO process to extract water from wastewater effluent is feasible.

### Integrating FO with AWE

A water-hydrogen balance model was established to quantify and balance the FO and AWE units in the FOWS$_{AWE}$ system, as shown in

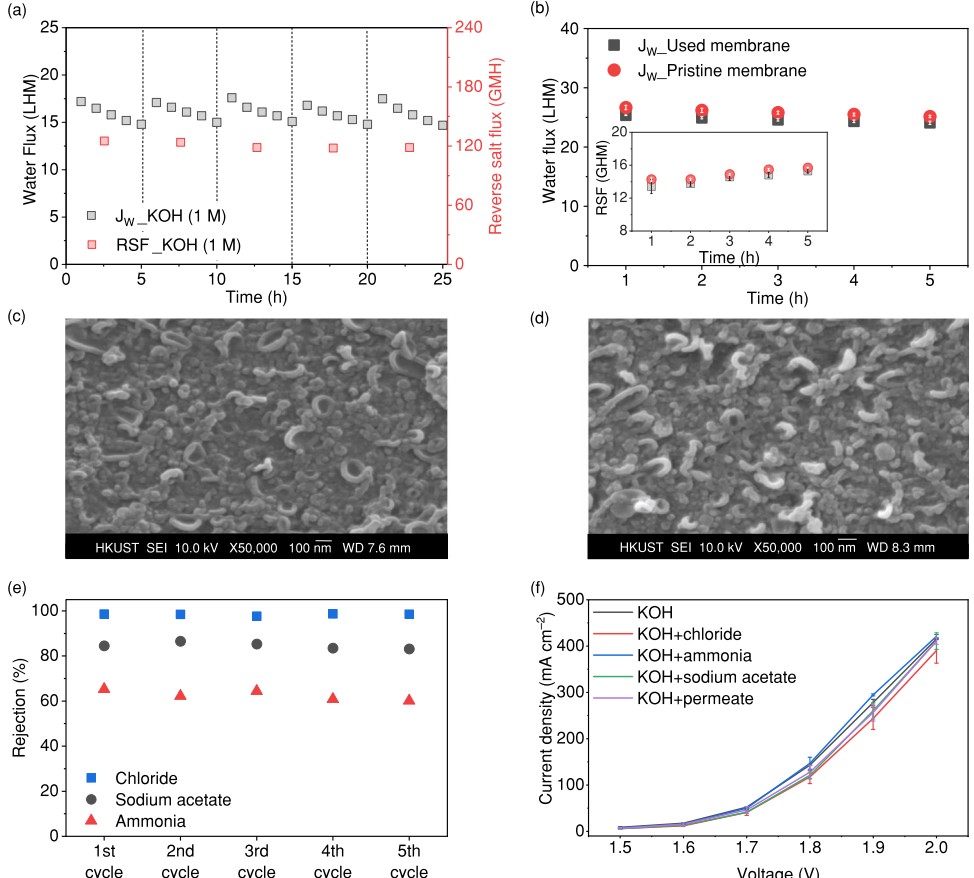

**Fig. 2 | Impacts of using KOH as a draw solution of FO processes to extract clean water from wastewater effluent. a** Water flux ($J_w$) and reverse salt flux (RSF) of the FO process with simulated wastewater effluent containing sodium acetate, NH$_3$/ NH$_4^+$, and chloride as feed solution and 1 M KOH as draw solution for five cycles. The dashed lines represent five cycles of 5 h each. (Conditions: [NaAc] = 7.5 mM, [NH$_3$/NH$_4^+$] = 2.2 mM as N, [Cl$^-$] = 4.2 mM, and flow rate = 70 ml min$^{-1}$); **b** comparison in water flux ($J_w$) and RSF between the pristine membrane and the 5-cycle used membrane using deionised water as feed solution and 1 M NaCl as draw solution. The inset figure shows the reverse salt flux from the two membranes (error bars represent standard deviation from two independent replicates); **c** SEM images showing the surface morphology of the selective layer on the pristine membranes; **d** SEM images showing the surface morphology of the selective layer on the 5-cycle used membrane; **e** the rejection of sodium acetate, ammonia, and chloride by the FO membrane using 1 M KOH as draw solution (conditions: [Sodium acetate] = 7.5 mM, [Ammonia] = 2.2 mM as N, [Chloride] = 4.2 mM, and flow rate = 70 ml min$^{-1}$); **f** J–V curves for 1 M KOH and 1 M KOH mixed with different impurities including sodium acetate, ammonia, and chloride (error bars represent standard deviation from three independent replicates). Source data are provided as a Source Data file.

Fig. 3a, with the objective of continuous operation to effectively produce H$_2$ from wastewater effluent. In the integrated system, clean water is first extracted from the wastewater effluent by the FO process using KOH as the draw solution and then split into H$_2$ and oxygen (O$_2$) by AWE using the 5-cell alkaline stack containing nickel alloy powder combined with nickel foam electrodes separated by polymer diaphragms (detailed setup is shown in the 'Methods' section). A sustainable operation of this system thus requires a dynamic equilibrium that enables the water production rate crossing the FO membrane ($Q_{FO}$) to match the water rate consumed by AWE ($Q_{AWE}$) as shown in Eq. 1, while maintaining a consistent concentration of KOH.

$$Q_{FO} = S\Delta C \frac{K_w I}{d} RT = Q_{AWE} = i_{cell} \frac{FE \cdot N_{cell} V_m}{2F} \quad (1)$$

$Q_{FO}$ is a function of controllable parameters in the FO process including the membrane area ($S$, cm$^2$) and the concentration gradient between draw solution and feed solution ($\Delta C$, M)[40]. The membrane permeability coefficient ($K_w$, L atm$^{-1}$ s$^{-1}$ cm$^{-1}$), when multiplied by the van't Hoff factor of the feed solution ($I$), represents how easily water can be extracted through the membrane, remaining constant throughout the test. Other parameters, including the membrane thickness ($d$ = 0.1 mm), the ideal

gas constant ($R$ = 0.0821 L atm K$^{-1}$ mol$^{-1}$), and temperature ($T$ = 296 K), also remained constant during the test. $Q_{AWE}$ is only controlled by the applied current of AWE ($i_{cell}$, A)[41], as other parameters, including the Faradaic efficiency (FE), the number of cells constituting the electrolyser stack ($N_{cell}$ = 5 for our setup), $V_m$ (as the molar volume of H$_2$O at 0.018 L mol$^{-1}$ at 296 K of $T$ and 1 atm), and Faraday's constant ($F$ = 96,485 C mol$^{-1}$) are constant throughout the test. Therefore, through balancing $Q_{FO}$ and $Q_{AWE}$, a linear correlation was established to describe the specific normalised current of the FOWS$_{AWE}$ system ($J_i$, A cm$^{-2}$), which is defined as the required current of this integrated system for handling the permeated water per area of FO membranes ($i_{cell}$ $S^{-1}$), changed with $\Delta C$ as shown in Eq. 2.

$$J_i = \frac{i_{cell}}{S} = \left( \frac{K_w I}{d} RT \frac{2F}{FE N_{cell} V_m} \right) \Delta C = \Phi \cdot \Delta C \quad (2)$$

where $\Phi$ is the H$_2$ production potential (L A mol$^{-1}$ cm$^{-2}$), determined by the configurations of the FO membranes and AWE, types of feed and draw solutions, and operational conditions. H$_2$ production potential ($\Phi$) was calculated by substituting the values of $K_w \times I$, $d$, $R$, $T$, FE, $N_{cell}$, and $V_m$ into Eq. 2, and $K_w \times I$ was obtained experimentally by measuring $Q_{FO}$ under given conditions of $S$ and $\Delta C$ in an independent FO module

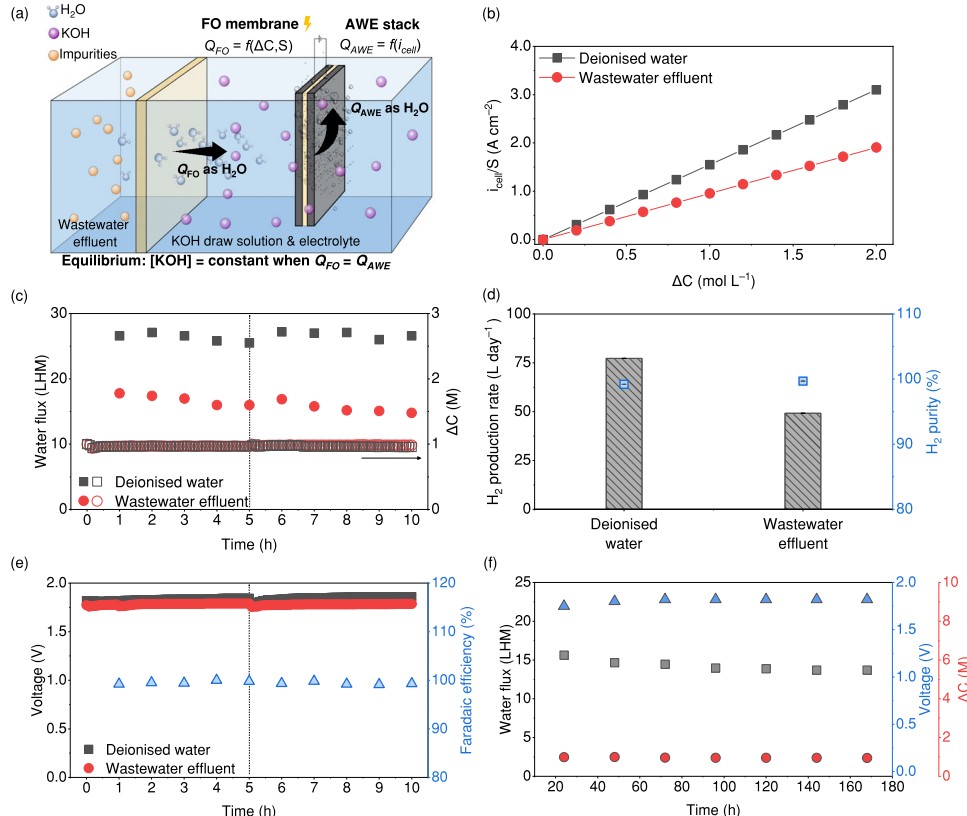

**Fig. 3 | Integrating the FO processes with alkaline water electrolysis (AWE).**
**a** Schematics showing a dynamic equilibrium to balance the water production rate crossing the FO membrane ($Q_{FO}$) and the water consumption rate by AWE ($Q_{AWE}$) in the FOWS$_{AWE}$ system; **b** the specific normalised current ($i_{cell}/S$) as a function of the concentration gradient ($\Delta C$) to reach the equilibrium in the FOWS$_{AWE}$ system fed with deionised water and wastewater effluent. Data can be reproduced using Eq. 2; **c** the water flux and $\Delta C$ of the FOWS$_{AWE}$ system during 2-cycle operation. The vertical dashed line represents the transition between the first cycle and second cycle; **d** the H$_2$ production rate and H$_2$ purity for using deionised water and

wastewater effluent as the feed of the FOWS$_{AWE}$ system (error bars represent standard deviation from three independent replicates); **e** the resulting voltage during the operation of the FOWS$_{AWE}$ system. The vertical dashed line represents the transition between the first cycle and the second cycle (conditions: [KOH] = 1 M, $S$ = 1 cm$^2$, $i_{cell}$ = 1.55 A for deionised water and 0.95 A for wastewater effluent (HK-WWE1)); **f** the water flux, voltage, and $\Delta C$ during the 168-h continuous operation of the integrated FOWS$_{AWE}$ system (conditions: [KOH] = 1 M, $S$ = 1 cm$^2$, and $i_{cell}$ = 0.90 A using the HK-WWE-2). Source data are provided as a Source Data file.

using Eq. 1, whereas FE was calculated by measuring the H$_2$ production rate and purity at a given $i_{cell}$ of the AWE module (detailed in the 'Methods' section). In this study, $\varPhi$ values for feeding deionised water and the wastewater effluent collected from a wastewater treatment plant in Hong Kong (HK-WWE1) to the FOWS$_{AWE}$ system were calculated as 1.55 and 0.95 A mol$^{-1}$ cm$^{-2}$, respectively. The lower $\varPhi$ for wastewater effluent was attributed to impurities in the wastewater effluent reducing $K_w \times I$. The obtained $\varPhi$ values were then used to generate the relationship between $J_i$ and $\Delta C$ when reaching a dynamic equilibrium for the FO and AWE units in the FOWS$_{AWE}$ system, as shown in Fig. 3b. Therefore, to maintain $\Delta C$ at 1 M, $J_i$ was calculated as 1.55 and 0.95 A cm$^{-2}$ for deionised water and HK-WWE1, respectively[21].

The two setups of FO and AWE were then experimentally integrated using a 1 cm$^2$ FO membrane and applying an $i_{cell}$ of AWE of 1.55 and 0.95 A for deionised water and HK-WWE1 according to their $J_i$ values, respectively (integrated process shown in Supplementary Fig. 17). As shown in Fig. 3c, d, the continuous operation of this integrated system over two cycles of 5 h each, demonstrated a stable water flux crossing the FO membrane, constant $\Delta C$, and a stable H$_2$ production rate when fed with either deionised water or secondary wastewater treated effluent. In addition, the molar ratio of the H$_2$ generation rate to $Q_{FO}$ was close to one after the two test cycles (Supplementary Fig. 7), indicating that nearly all the water produced from FO was consumed by AWE. These observations indicate a successful integration of FO and AWE processes and validate our

established dynamic equilibrium for the FOWS$_{AWE}$ system. Using wastewater effluent as the feed resulted in a 38% decrease in both $Q_{FO}$ and H$_2$ production rate compared to feeding deionised water, which aligns with the decrease observed in the value of $K_w \times I$ from wastewater effluent. This indicates that these impurities mainly affect the capacity of the water produced from FO, and the ability of the FOWS$_{AWE}$ system to produce H$_2$ relies on this water production capacity. Nevertheless, using wastewater effluent as the feed did not compromise the H$_2$ purity, as evidenced by the comparable H$_2$ percentage observed in samples produced using both deionised water and wastewater effluent, as shown in Fig. 3d. Additionally, the stability of H$_2$ production was not affected either, as the FOWS$_{AWE}$ system operated at a near-constant voltage of 1.79 ± 0.01 V and maintained a Faradaic efficiency of over 99% throughout two cycles of 5 h each (Fig. 3e), indicating that there were no side reactions caused by impurities crossing the FO membranes or accumulation of undesired impurities.

A continuous 168-h operation of the FOWS$_{AWE}$ system was conducted to assess its long-term stability under consistent conditions using a wastewater effluent sample collected from a second wastewater treatment plant in Hong Kong (HK-WWE2), and the result is shown in Fig. 3f. The water flux decreased by -12% from 15.6 to 13.7 LHM, whereas the voltage increased by 4% during the initial 24 h, starting from 1.75 V and maintaining at 1.82 V. In addition, the system sustained a $\Delta C$ consistently at around 1 M during the entire 168-h

testing period. The 12% reduced water flux after the long-term operation was attributed to fouling from organic matter in the wastewater[42], because the ratio of C to O elements on the membrane surface was reduced by ~13%, and no degradation of integrity for both the selective and support layers of the FO membrane was observed in SEM coupled with Energy Dispersive X-ray (Supplementary Table 3 and Fig. 8, respectively). FO membrane fouling was shown to be largely reversible, unlike the often irreversible fouling challenges faced in RO[43], which means that the reduced water flux in our system can be easily recovered by backwashing or chemical washing. Notably, the $H_2$ gas purity remained stably high at over 99% throughout the entire long-term operation (Supplementary Fig. 9). These results demonstrated the operational stability of the FOWS$_{AWE}$ system throughout the 168-h testing period. In addition, as the FOWS$_{AWE}$ system relies on the water flux to generate $H_2$, we tested more wastewater samples from the southern and northeastern regions of China, with their composition and concentrations detailed in Supplementary Table 4. The results show that our system exhibited a constant water flux, voltage, and $\Delta C$ for various impurities and pH values from different wastewater samples (Supplementary Figs. 10–12). Despite the varied $K_w \times I$ caused by different impurities, the resulting $H_2$ production rates were consistent and remained within a range of 5% (Supplementary Fig. 11d). These results validate the stability and adaptability of our system for producing $H_2$ under varying wastewater conditions.

## $H_2$ production and energy efficiency of the FOWS$_{AWE}$ system

The $H_2$ production and energy efficiency of the FOWS$_{AWE}$ system showed superior performance compared with two other membrane-based P2H systems for seawater purification integrated with electrolytic water splitting: (i) the seawater electrolysis system (SES) using hydrophobic PTFE membranes with 30% KOH as a self-dampening electrolyte (SDE), and (ii) the FOWS employing cellulose triacetate (CTA) membranes with 0.8 M sodium phosphate as the electrolyte[21,22]. The $H_2$ produced in the three systems was compared by normalising the $H_2$ production rate over the area of water-producing membranes, as shown in Fig. 4a. The FOWS$_{AWE}$ system produced the highest $H_2$ rate at 448 Nm³ day⁻¹ m⁻² of TFC-FO membrane, which is 102 and 14 times higher than those of SES and FOWS, respectively. This significant improvement in $H_2$ production compared with SES is primarily due to the higher $\Phi$ provided by FO membranes in producing more water. Unlike SES using hydrophobic PTFE membranes, the TFC-FO membrane is hydrophilic and thus offers higher $K_w$ to allow water to permeate. This is verified by the 23–32 times higher $J_i$ of the TFC-FO membrane to reach equilibrium than the PTFE membrane at the same values of $S$, $\Delta C$, and using identical AWE setups (Supplementary Fig. 13). Besides, PTFE membranes face wetting issues during operation, which reduces the ability to reject impurities as water penetrates the pores[44,45]. The enhanced $H_2$ production compared to FOWS is mainly attributed to the 5× higher $\Delta C$ between wastewater effluent and 1 M KOH compared to the 0.2 M $\Delta C$ between seawater and 0.8 M sodium phosphate, enhancing the driving force for producing water. The $H_2$ flux comparison along with the water-hydrogen model suggests the water production rate dominates in determining the $H_2$ flux in a membrane-electrolysis system, highlighting the importance of optimising both the membrane type and the feed water source. Moreover, high water production rates offer additional benefits of requiring fewer membranes and associated equipment, thereby reducing space and capital costs which are particularly important considerations for scaling up the system.

The FOWS$_{AWE}$ system exhibits low energy consumption, as shown in Fig. 4b, where the SEC of the three systems is compared in addition to direct wastewater electrolysis. The FOWS$_{AWE}$ system requires the lowest SEC at 4.43 kWh Nm⁻³ at an $i_{cell}$ of 0.95 A using wastewater effluent as feed. This SEC is comparable to commercial alkaline electrolysis systems that use deionised water and is 10%, 34%, and 84% lower than SES, FOWS, and direct electrolysis of wastewater, respectively[46]. The lower SEC highlights the importance of preventing undesired impurities from electrolysis and using a pH-tolerant membrane to enable the use of KOH as an electrolyte as it has higher ionic conductivity for more efficient electron transfer. Such excellent energy efficiency is also applicable to other commercial AWE configurations, if the applied current matches the water-hydrogen equilibrium, making the FOWS$_{AWE}$ system easily adaptable and scalable to meet varying market demands. Therefore, the hydrophilic property of TFC-FO membranes, the higher $\Delta C$ resulting from wastewater effluent usage instead of seawater, and the use of an optimal electrolyte (i.e.

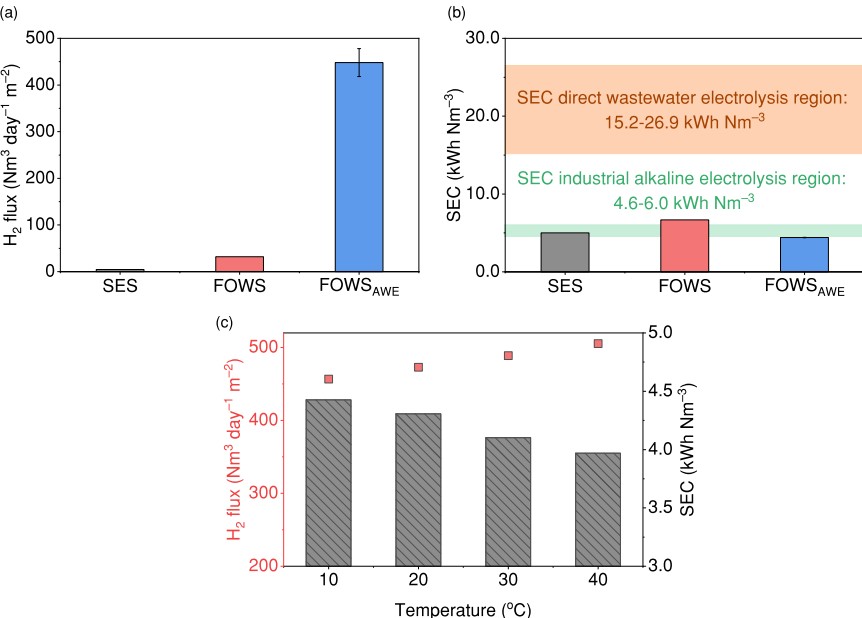

**Fig. 4 | $H_2$ production and energy efficiency of the FOWS$_{AWE}$ system. a** The comparison in $H_2$ flux among three membrane-based P2H systems (error bar represents the standard deviation from three independent replicates); **b** the comparison in specific energy consumption (SEC) among three membrane-based P2H systems. The orange region represents the SEC range typical for direct wastewater electrolysis. The green region represents the SEC range for industrial alkaline water electrolysis; **c** $H_2$ flux and SEC of the FOWS$_{AWE}$ system changed with KOH temperatures. Source data are provided as a Source Data file.

KOH) all contribute to a high $H_2$ flux and energy-efficient P2H conversion.

In addition, electrolysis is an exothermic process that generates heat, so capturing and utilising waste heat during AWE operations is critical in achieving the energy-efficient operation of the system. By simulating the capture and utilisation of the waste heat during AWE operations to heat up the KOH draw solute temperature to 40 °C, the FOWS$_{AWE}$ system shows an increase of $H_2$ flux by 11% and a decrease of SEC from 4.43 to 3.96 kWh Nm$^{-3}$ (Fig. 4c and detailed calculations provided in Supplementary Note 1). This brings us closer to the target of 3.75 kWh Nm$^{-3}$ proposed by the International Renewable Energy Agency[47]. The enhanced $H_2$ flux is mainly due to the increased water production at higher temperatures, indicated by Eq.1[48], while the reduced SEC is attributed to the increased KOH ionic conductivity reducing the overall resistance of the cell and enabling more efficient electron transfer (Supplementary Fig. 14). These results also suggest that the FOWS$_{AWE}$ system can perform better in hot regions where the wastewater has temperatures of up to 35 °C and can potentially heat the draw solutions during membrane operational processes[28], or in areas with abundant solar energy that can be used for heating.

### Implications for $H_2$ production using wastewater effluent

This study demonstrated the use of municipal wastewater effluent by a FOWS$_{AWE}$ system for fast and energy-efficient green $H_2$ generation to address the large-scale $H_2$ production need associated with potential water constraints. The FOWS$_{AWE}$ system exhibits stable and continuous $H_2$ production at the highest yields of $H_2$ to date (448 Nm$^3$ day$^{-1}$ m$^{-2}$) using alternative water resources achieving a high $H_2$ purity of >99% and low SEC of 4.43 kWh m$^{-3}$ during the entire testing period of over 10 h. In addition, the FOWS$_{AWE}$ system demonstrated long-term stability by being operated continuously for 168 h using real wastewater effluent and proved its capability and adaptability in producing $H_2$ across diverse wastewater conditions found in different regions.

Using wastewater effluent is crucial for sustainable $H_2$ production in regions where seawater is unavailable or freshwater resources are scarce, especially as $H_2$ projects are rapidly expanding amid the foreseeable intensified worldwide water stress. The modular FO and AWE system along with the established water-hydrogen balance model allow the FOWS$_{AWE}$ system to operate with different influent water quality at different scales, ensuring an easy yet versatile approach for sustainable P2H conversion from the household to the city scale. The integration of TFC-FO with electrolysis can reduce the capital costs of water treatment by up to 46% compared to conventional RO and FO–RO processes[49], providing a strong economic incentive for FOWS$_{AWE}$ adoption. Addressing pressing global challenges such as water scarcity, climate change, and energy insecurity requires urgent action. Existing technologies, such as the FO and AWE, can substantially impact on solving these challenges if applied and integrated more effectively. The FOWS$_{AWE}$ system exemplifies leveraging synergistic technologies to rapidly develop economical, efficient, and sustainable solutions for interconnected environmental issues.

In addition to the $H_2$ production market for industrial use and energy storage, the FOWS$_{AWE}$ system provides advantages to wastewater treatment plants and many industries, as it serves as a tool for water reuse. The system has the potential to reduce the treatment load and wastewater discharge while simultaneously generating onsite energy, and mitigating environmental impacts[50]. It is estimated that an onsite FOWS$_{AWE}$ station with a capacity of 5–6 MW can produce $O_2$ at ~20,000 kg day$^{-1}$, which can supply the aerobic processes of a wastewater treatment plant in treating 50,000 m$^3$ day$^{-1}$ of wastewater (calculation shown in Supplementary Note 2)[51,52] or support high-density in-land and offshore aquaculture[53,54]. Moreover, fuel cells that use $H_2$ to generate electricity also produce high-purity water as a byproduct,

which can be supplied to industries that require it, e.g. semiconductor and pharmaceutical industries.

## Methods
### Chemicals

Potassium hydroxide (KOH, Scharlau, 85.0–100.5%), sodium bicarbonate (NaHCO$_3$, Sigma-Aldrich, 99.5–100.5%), potassium pyrophosphate (K$_4$P$_2$O$_7$, Macklin, 99%), sodium chloride (NaCl, Sigma-Aldrich, ≥99%)), ammonium chloride (NH$_4$Cl, Sigma-Aldrich, ≥99.5%), and sodium acetate (CH$_3$COONa, Sigma Aldrich, ≥99%). The stock solutions were prepared by dissolving these chemicals in deionised water (18.2 MΩ cm) produced by a water purification system (Millipore, USA). The simulated wastewater effluent was prepared by adding 7.5 mM of sodium acetate, 2.7 mM of NH$_3$/NH$_4^+$, and 4.2 mM of chloride into deionised water. Two wastewater effluent samples were collected from two different treatment plants in Hong Kong SAR labelled as HK-WWE1 and HK-WWE2. Additionally, one wastewater effluent sample was obtained from Guangdong Province, China, designated as GD-WWE, and another from Northeast China, referred to as NC-WWE. All the samples were filtered with high-pressure filtration equipment using cellulose membrane filters (pore size: 0.45 μm) and stored at 4 °C. The quality of the wastewater effluent samples used in this study are presented in the Supplementary Table 4.

### FO to extract clean water

The experiments to produce clean water from FO processes were conducted in an independent FO module as shown in Supplementary Fig. 15. The FO module consists of an FO cell with two chambers with an area of 12 cm$^2$ and a depth of 0.3 cm separated by a single flat FO membrane sheet (051303, Porifera). The feed solution chamber was connected to a feed solution tank with a volume of 500 mL while another was connected to a draw solution tank with a volume of 200 mL. The feed solution was recirculated from the feed tank to the FO feed solution chamber by a peristaltic pump at a flow rate of 70 mL min$^{-1}$. On the other side, the draw solution was recirculated from the draw solution tank to the FO draw solution chamber at the same flow rate. The water flux ($J_W$, LHM) was gravimetrically measured by the loss of water weight in the feed tank per unit of time using an electronic balance (FX-3000GD, A&D Instruments) divided by the membrane area (Eq. 3).

$$J_W(LHM) = \frac{\Delta M}{\rho \cdot S \cdot \Delta t} \tag{3}$$

where $\Delta M$ is the water weight change during the test (g), $\rho$ is the water density of 1 kg L$^{-1}$, $S$ is the membrane area (m$^2$), and $\Delta t$ is the time interval of recording water weight (h). The RSF was determined by measuring the potassium ion concentration via an Inductively Coupled Plasma-Optical Emission Spectrometer (ICP-OES, 725-ES, Varian). The RSF ($J_S$, GHM) was obtained by the changes in KOH concentration to the feed solution, as shown in Eq. 4.

$$RSF(GHM) = \frac{V_t \cdot C_t - V_0 \cdot C_0}{S \cdot \Delta t} \tag{4}$$

where $V_0$ and $V_t$ represent the volumes of the feed solution before and after testing, while $C_0$ and $C_t$ represent the concentrations of KOH in the feed solution before and after the tests, respectively. The conductivity and the pH of the feed solution were monitored using a multiparameter benchtop metre (Multi 9630 IDS, WTW inoLab). The rejection of impurities by the FO membrane was calculated using Eq. 5:

$$\text{Rejection}(\%) = \left(1 - \frac{C_P}{C_f}\right) \times 100 \tag{5}$$

where $C_p$ and $C_f$ represent the concentrations of the permeated impurities including sodium acetate, $NH_3/NH_4^+$, or chloride through FO membrane and their initial feed concentrations, respectively. The concentrations of sodium acetate and $NH_3/NH_4^+$ were measured by a total organic carbon analyser coupled with a total nitrogen measurement (TOC-L analyser with TNM-L, Shimadzu). The concentrations of $NH_3/NH_4^+$ in the wastewater effluent samples were measured by spectrophotometric method. The concentrations of chloride were quantified using ion chromatography (IC, 940 Professional IC Vario, Metrohm). The membrane before and after use was characterised by SEM (JSM-6700F, JEOL and MAIA3, Tescan) and FTIR (Vertex 70, Bruker).

### Water electrolysis system to produce hydrogen
The experiments to produce $H_2$ were conducted in an independent AWE module, as shown in Supplementary Fig. 16. The AWE module consists of a 5-cell alkaline stack (LBE-5SC, Light Bridge Inc.) that utilises bipolar electrodes separated by polymer diaphragms, an electrolyte supply tank, and a gas separator for collecting $H_2$ and $O_2$. The electrodes, that are composed of nickel alloy powder combined with nickel foam, have a thickness of 0.8 mm, a diameter of 44 mm, an area of $15.2\ cm^2$, and a mass loading of $74.3\ mg\ cm^{-2}$. Meanwhile, the polymer separators have a thickness of 100–130 μm, a diameter of 51 mm, and an area of $15.2\ cm^2$. The lower potential cut-off was determined to be 1.5 V per cell. The alkaline stack was supplied with different electrolytes using a peristaltic pump. The J–V curves of different electrolytes were collected using a DC power supply (DP3030, MESTEK). The impact of impurities on water electrolysis was determined by adding the target compounds in the electrolyte to obtain J–V curves. The impact of temperature on water electrolysis was obtained by regulating the electrolyte temperature using a water bath (HH-1, KOY) in the AWE module for generating J–V curves.

### Integrating FO and AWE to make the $FOWS_{AWE}$ system
The process design of the $FOWS_{AWE}$ system is shown in Supplementary Fig. 17. Clean water was first extracted from the feed wastewater effluent tank via the FO process with the same setup as the independent FO module described earlier except the membrane area was $1\ cm^2$. The extracted clean water diluted the KOH draw solution whose concentration was then recovered by splitting the water in the AWE using the diluted KOH as the electrolyte. The AWE outflow consisted of two KOH streams, one mixed with $H_2$ and the other mixed with $O_2$. The $H_2$ and $O_2$ were then separated from the KOH streams using a gravity-based gas–liquid separator, and the KOH stream free of gases flowed back to the draw solution tank. The KOH concentration was continuously monitored by analysing the change in the conductivity of the draw solution and the change in the pH of feed the wastewater effluent. $\Delta C$ was determined by subtracting the molar concentration of all impurities in the feed tank ($\Sigma[impurities]_{feed}$) as listed in Supplementary Table 4 and the molar concentration of KOH in the feed tank due to RSF ($[KOH]_{feed}$) from the molar concentration of recovered KOH in the draw solution tank ($[KOH]_{recovered}$) as shown in Eq. 6:

$$\Delta C = [KOH]_{recovered} - \Sigma[impurities]_{feed} - [KOH]_{feed} \tag{6}$$

The separated $H_2$ was collected by draining the water downwards in an inverted measuring cylinder, and the $H_2$ production rate was determined by the water drainage volume over pre-determined time intervals. The $H_2$ flux was then calculated by dividing the obtained $H_2$ production rate by the FO membrane areas. The purity of the collected $H_2$ was analysed by gas chromatography (990 Micro GC System, Agilent) equipped with a thermal conductivity detector. The obtained $H_2$ production rates and $H_2$ purity were used to determine the Faradaic efficiency (FE) of the system by dividing the experimentally measured $H_2$ production rate ($r_{H_2}$) by the theoretical

$H_2$ production rate ($r_{H_2,ideal}$) (Eq. 7).

$$FE_{H_2} = \frac{r_{H_2}}{r_{H_2,ideal}} = \frac{r_{H_2}}{\frac{i}{2F} \times \frac{RT}{P_0}} \tag{7}$$

where $P_0$ is atmospheric pressure (101 kPa), and $T$ is the operating temperature (296 K). The SEC of the $FOWS_{AWE}$ system (kWh Nm$^{-3}$) was calculated using Eq. 8.

$$SEC = \frac{i_{cell} \times U}{r_{H_2}} \tag{8}$$

where $U$ is the resulting voltage (V) obtained. The $i_{cell}$ value was 1.55 A when deionised water was used as the feed source, 0.95 A when HK-WWE-1 was used as the feed source, and 0.90 A when HK-WWE-2 was used as the feed source during the long-term test.

## Data availability
The data that supports the findings of the study are included in the main text and supplementary information files. Source data are provided with this paper.

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

## Acknowledgements
This work was partially funded by the Hong Kong Research Grants Council (T21-604/19-R, C.S.), the Fundamental Research Funds for the Central Universities (310400209521, L.L.), HKUST 30 for 30 Research Initiative Scheme (3030_010, C.S.), the National Science Foundation (EEC-1449500, P.W.) Nanosystems Engineering Research Center on Nanotechnology-Enabled Water Treatment, and the Talent Startup Fund of Beijing Normal University (310432104, L.L. and 312200502503, L.L.).

## Author contributions
G.S.C., C.S., Y.S. and L.L. conceived the idea and designed the analysis. G.S.C., N.K.K., A.M. and Y.S., conducted the experiments. G.S.C., C.S., N.K.K., A.K.A., Y.S., P.W. and L.L. analysed the data. G.S.C., C.S., A.K.A., N.K.K., F.C., A.M., P.W., Y.S. and L.L. wrote and revised the manuscript. Y.S. and L.L. coordinated and supervised the research.

## Competing interests
The authors declare no competing interests.

## Additional information

[1]Department of Civil and Environmental Engineering, The Hong Kong University of Science and Technology, Hong Kong SAR, China. [2]Hong Kong Branch of Chinese National Engineering Research Center for Control & Treatment of Heavy Metal Pollution, The Hong Kong University of Science and Technology, Hong Kong SAR, China. [3]School of Energy and Environment, City University of Hong Kong, Hong Kong SAR, China. [4]NYUAD Water Research Center, New York University Abu Dhabi, Abu Dhabi, United Arab Emirates. [5]Department of Mechanical and Aerospace Engineering, The Hong Kong University of Science and Technology, Hong Kong SAR, China. [6]Chair of Electrode Design for Electrochemical Energy Systems, University of Bayreuth, Bayreuth, Germany. [7]Nanosystems Engineering Research Center for Nanotechnology-Enabled Water Treatment, School of Sustainable Engineering and The Built Environment, Arizona State University, Tempe, AZ, USA. [8]Advanced Interdisciplinary Institute of Environment and Ecology, Beijing Normal University, Zhuhai, China.
✉ e-mail: ysongat@connect.ust.hk; lling@bnu.edu.cn

