## [Peer Review File · Nature Communications]

Ultra-fast green hydrogen production from municipal wastewater by an integrated forward osmosis-alkaline water electrolysis systemREVIEWER COMMENTS

Reviewer #1 (Remarks to the Author):

Cassol et al. describe an electrochemical system where forward osmosis is coupled to water splitting in basic conditions to produce hydrogen gas from wastewater. This work expands upon the original discovery of the forward osmosis water-splitting (FOWS) system (Proc. Natl. Acad. Sci. USA 2021, 118, e2024855118) such a system by employing a potassium hydroxide-tolerant membrane, allowing for more energy-efficient electrochemistry. The authors employ a range of electrochemical (e.g., cyclic voltammetry, chronopotentiometry) and analytical techniques (e.g., electron microscopy, gas chromatography) to validate the selectivity and stability of forward osmosis and water splitting processes. This reviewer finds the technical quality of this work high and appreciates the thoroughness with which Cassol et al. designed and assessed their device, as well as their techno-economic analyses.

I went and read the original paper by Nocera and Veroneau in PNAS. I have to say that the present authors designed a conceptually identical system that performs water splitting in base. While this is an important improvement of the device previously proposed, it is not the novel system as the authors suggest. Indeed, in the original FOES paper it is plainly stated "Water splitting in base or acid solutions will greatly improve the energy efficiency but ion-selective membranes that are stable to concentrated base or acid will be required." Thus, from this perspective, this is not a forward osmosis-alkaline water electrolysis (FO-AWE) system. This is an alkaline FOWS system. To suggest otherwise is to mislead readers and to add an unnecessary acronym to the literature.

In summary, this reviewer finds the work technically sound and a significant enough advancement for publication in Nature Communications; however, this would be contingent on describing their device in a straightforward way as an alkaline FOWS system. There is no meaningful difference between this device and that previously described by Veroneau and Nocera, particularly as the authors use a commercially available electrochemical cell.

Minor Comments

- Around Line 70, we encourage the authors to highlight the issue of precipitation at the cathode, which is more immediately detrimental to hydrogen gas production than chloride oxidation.
- Around Line 85, we encourage the authors to motivate basic conditions over acidic ones, which might be similarly advantageous for hydrogen gas production.
- On Line 97, remove the qualifier 'novel'.
- On Line 101, 'allows to' should be 'allowing the'.
- On Line 217, please refer to Ref. 21 again as these calculations are also described therein.
- Please reiterate in the main text what your anode and cathode materials were.
- On Line 297, please clarify why these long-term operational studies were not performed.

Reviewer #2 (Remarks to the Author):

Developing membrane-based water electrolysis system which can be used for different water source is important for using renewable energy to produce green hydrogen. This manuscript submitted by Ling et al. reports a modularized forward osmosis-alkaline water electrolysis system to allow for using wastewater to generate high pure hydrogen. The authors suggest that the integrated water electrolysis system enables to use 1 M potassium hydroxide as a draw solution and electrolyte to split water and produce hydrogen. Moreover, the water-hydrogen balance model was established to balance the production and consumption of water based on the reasonable calculations. The integrated system exhibited an excellent H₂ production rate of 448 Nm³ day⁻¹ 99 per m² of membrane area with energy consumption as low as 3.96 kWh Nm⁻³. This work provides a promising strategy for energy-efficient hydrogen generation in areas where water resources are scarce and seawater is unavailable. Therefore, I recommend to publish this manuscript after minor revisions. Below is a list of specific comments and questions the authors should address before publication.

1. In this manuscript, the permeate and rejection tests were performed in 1 M KOH mixture consisting of NaAc, NH₃/NH₄⁺ and chloride. In fact, some impurities including phosphorous, trace metal ions and organics also containing in wastewater, which may have an effect on the extraction of water from wastewater. Therefore, the experiments need to be performed to evaluate the impact of above impurities on the TFC-FO membranes and the water flux.
2. The pH of wastewater also has an effect on TFC-FO membranes, the water flux, and the hydrogen production. Please supply the permeate and rejection tests in wastewater with different pH.
3. Long-term durability is an important indicator to evaluate the potential for practical application of the water electrolysis system. The stability test for the longer time is necessary in this work.
4. The species and content of impurities in wastewater vary in different regions. In order to make this integrated water electrolysis system more convincing, the H₂ generation rate tests need to be performed in the wastewater effluent sample which are collected from the different regions.
5. There are some vague description and typo errors in this article. Thus, for a better readability, please check throughout the whole article text and correct all the errors.

Responses to comments of Reviewer #1:

Cassol et al. describe an electrochemical system where forward osmosis is coupled to water splitting in basic conditions to produce hydrogen gas from wastewater. This work expands upon the original discovery of the forward osmosis water-splitting (FOWS) system (Proc. Natl. Acad. Sci. USA 2021, 118, e2024855118) such a system by employing a potassium hydroxide-tolerant membrane, allowing for more energy-efficient electrochemistry. The authors employ a range of electrochemical (e.g., cyclic voltammetry, chronopotentiometry) and analytical techniques (e.g., electron microscopy, gas chromatography) to validate the selectivity and stability of forward osmosis and water splitting processes. This reviewer finds the technical quality of this work high and appreciates the thoroughness with which Cassol et al. designed and assessed their device, as well as their techno-economic analyses.

I went and read the original paper by Nocera and Veroneau in PNAS. I have to say that the present authors designed a conceptually identical system that performs water splitting in base. While this is an important improvement of the device previously proposed, it is not the novel system as the authors suggest. Indeed, in the original FOES paper it is plainly stated "Water splitting in base or acid solutions will greatly improve the energy efficiency but ion-selective membranes that are stable to concentrated base or acid will be required." Thus, from this perspective, this is not a forward osmosis-alkaline water electrolysis (FO-AWE) system. This is an alkaline FOWS system. To suggest otherwise is to mislead readers and to add an unnecessary acronym to the literature.

In summary, this reviewer finds the work technically sound and a significant enough advancement for publication in Nature Communications; however, this would be contingent on describing their device in a straightforward way as an alkaline FOWS system. There is no meaningful difference between this device and that previously described by Veroneau and Nocera, particularly as the authors use a commercially available electrochemical cell.

Reply: We appreciate the comments from the reviewer. Strictly speaking, our system is indeed an advanced version of the FOWS concept, designed to operate more effectively with alkaline conditions. By carefully selecting KOH as electrolyte and verifying its effectiveness in extracting water in FO with a KOH-tolerant TFC membrane, we identified the optimal compatibility between FO and water electrolysis, thus producing a 14-fold increase in H₂ flux while consuming 34% less

energy. Furthermore, we have verified for the first time that the system we designed works effectively with wastewater at a high faradaic efficiency of over 99%, increasing the water flux due to a higher concentration gradient and reducing the harmful chlorine by-products caused by abundant Cl^- when using seawater, and offering a valuable option for inland communities with limited water resources. We also demonstrated simple integration of two mature systems into a modular design that is both flexible and scalable, sidestepping the need for a more intricate FOWS cell architecture.

We agree with the reviewer to keep the original term of “FOWS” to avoid any misunderstanding. Meanwhile, we would like to point out that our integrated system is a specific design of FOWS which was operated under alkaline conditions for maximizing efficiency. We changed the name of our integrated system from “FO-AWE” to “FOWS_{AWE}” to emphasize our system for water splitting is using a specific technology, i.e., alkaline electrolysis, because "water splitting" could otherwise relate to other techniques like proton exchange membrane (PEM) or anion exchange membrane (AEM) electrolysis. "FO-AWE" has been changed to "FOWS_{AWE}" throughout the revised manuscript.

We also toned down the claim of novelty and updated the introduction and results sections to better reflect and give due credit to the original FOWS study. Further, we put more emphasis in our designed system towards ideal membrane-based electrolysis that allows for fast, sustainable, and economically viable H_2 production and waste recovery applications not limited by regional locations, contributing to a decarbonised future.

Comment 1:

Around Line 70, we encourage the authors to highlight the issue of precipitation at the cathode, which is more immediately detrimental to hydrogen gas production than chloride oxidation.

Reply: We appreciate the reviewer for raising this important point. We have added further context about one of the key challenges of direct seawater electrolysis¹, as suggested.

These are revised from **the newly revised Manuscript:**

Former text in Lines 69–70: “The direct seawater electrolysis results in limited lifespan and low H_2 production rates due to corrosivity and parasitic chlorine by-product formation at the anode.” was revised as:

Lines 69–72 “Direct seawater electrolysis results in **limited electrolyser lifespan** and low H₂ production rates **due to various factors including the precipitation of cations such as calcium and magnesium at the cathode, electrode corrosion,** and the formation of parasitic chlorine by-products at the anode^{12,13,14,15}.”

Reference:

1. Becker, H. *et al.* Impact of impurities on water electrolysis: a review. *Sustain. Energy Fuels* **7**, 1565–1603 (2023).

Comment 2:

Around Line 85, we encourage the authors to motivate basic conditions over acidic ones, which might be similarly advantageous for hydrogen gas production.

Reply: We appreciate this valuable suggestion. Although water splitting can be accelerated in either acidic or alkaline media due to higher conductivity, as suggested by Nocera and Veroneau², the use of basic KOH as the draw solution and electrolyte presents various advantages over acidic media. Utilizing KOH in water electrolysis not only enables the use of less expensive and abundant materials such as nickel as electrodes and catalysts, but also improves system durability by reducing the corrosion in acidic media, particularly at the anodes where the highly oxidative environment facilitates the corrosion^{3,4}. Additionally, alkaline water electrolysis has higher technology readiness levels, and thus offers higher safety levels, reliability and scalability. Moreover, alkaline conditions tend to produce less chlorine compared to acidic conditions due to the higher overpotentials and slower kinetics required for the chlorine evolution reaction in the presence of chloride⁵. These advantages support the selection of KOH for developing our more sustainable and economically viable FOWS_{AWE} system.

These are added to **the newly revised Manuscript:**

Former text in Lines 83–86: “We hypothesize that integrating FO with alkaline water electrolysis (AWE) using potassium hydroxide (KOH) as their shared operational solute is a superior strategy, because KOH is highly conductive and optimal for H₂ production in AWE and may also be a good draw solution for FO to provide osmotic gradient to enhance water production.” was revised as:

Lines 87–94: “**We hypothesise that utilising potassium hydroxide (KOH) for FOWS is more efficient because KOH ensures high current densities to split the water via alkaline water**

electrolysis (AWE) and may serve as a suitable draw solution for FO to provide osmotic gradient to extract water if coupled with a KOH-compatible FO membrane. Although acidic media may also enhance FOWS, using KOH is preferred as it allows the use of cost-effective materials (i.e., nickel) and minimises chlorine by-products due to higher overpotential and slower chlorine evolution reaction (CER) kinetics^{24,25}. Further, the technological maturity of AWE compared to the acidified ones ensures superior safety, reliability, and scalability.”

References:

2. Veroneau, S. S. & Nocera, D. G. Continuous electrochemical water splitting from natural water sources via forward osmosis. *Proc. Natl. Acad. Sci.* **118**, (2021).
3. Lei, Z. *et al.* Recent Progress in Electrocatalysts for Acidic Water Oxidation. *Adv. Energy Mater.* **10**, (2020).
4. Hou, Y. *et al.* Strategies for Electrochemically Sustainable H₂ Production in Acid. *Adv. Sci.* **9**, (2022).
5. Fei, H. *et al.* Direct Seawater Electrolysis: From Catalyst Design to Device Applications. *Adv. Mater.* (2023) doi:10.1002/adma.202309211.

Comment 3:

On Line 97, remove the qualifier 'novel'.

Reply: We appreciate the reviewer's comment on this sentence, and we have removed the term "novel" for improved clarity in the revised manuscript.

Comment 4:

On Line 101, 'allows to' should be 'allowing the'.

Reply: We appreciate the reviewer's comment. We have modified the relevant sentence in the revised manuscript by changing "allows to" to "allowing the" for improved clarity and grammatical correctness.

Comment 5:

On Line 217, please refer to Ref. 21 again as these calculations are also described therein.

Reply: We appreciate the reviewer for pointing out the missing reference. As part of revising the manuscript in response to this feedback, we have added the relevant citation at the appropriate point in the text.

Comment 6:

Please reiterate in the main text what your anode and cathode materials were.

Reply: We thank the reviewer for this suggestion. In our experiments, we used nickel alloy-based electrodes as both the anode and cathode materials for the alkaline water electrolysis system. To address this comment, we have included the following statements in the revised manuscript:

Former text in Lines 165–170: “Their potential impacts on electrolysis were then investigated by comparing the current density-voltage (J-V) curves generated from:” was revised as:

Lines 174-177: “The potential impacts to the electrolysis process from the permeated impurities were then investigated using a commercial 5-cell alkaline stack with nickel alloy-based electrodes by comparing the current density-voltage (J-V) curves generated from:”

Former text in Lines 183–186: “In the integrated system, clean water is first extracted from wastewater effluent by the FO process using KOH as the draw solution, and then split into H₂ and oxygen by AWE comprising a commercial 5-cell alkaline stack with nickel alloy-based electrodes divided by polymer diaphragms (detailed setup shown in *Methods*).” was revised as:

Lines 192-196: “In the integrated system, clean water is first extracted from the wastewater effluent by the FO process using KOH as the draw solution, and then split into H₂ and oxygen (O₂) by AWE using the 5-cell alkaline stack containing nickel alloy powder combined with nickel foam electrodes separated by polymer diaphragms (detailed setup is shown in the *Methods* section).”

Former text in Lines 371–374: “The AWE module consists of a 5-cell alkaline stack (LBE-5SC, Light Bridge Inc.) equipped with series-connected bipolar nickel alloy-based electrodes (0.8 mm thickness and 12 cm² effective surface areas each) separated by polymer diaphragms, an electrolyte supply tank, and a gas separator for collecting H₂ and O₂.” was revised as:

Lines 409-412: “The AWE module consists of a 5-cell alkaline stack (LBE-5SC, Light Bridge Inc.) that utilises bipolar electrodes composed of nickel alloy powder combined with nickel foam (0.8

mm thickness and 12 cm² effective surface areas each) separated by polymer diaphragms, an electrolyte supply tank, and a gas separator for collecting H₂ and O₂.”

Comment 7:

On Line 297, please clarify why these long-term operational studies were not performed.

Reply: We appreciate the reviewer's comment and acknowledge the importance of long-term operational studies. In response, we conducted a 168-h continuous test using our integrated system fed with a real wastewater effluent obtained from a second Hong Kong wastewater treatment plant (HK-WWE2, its water qualities shown in Supplementary Table 4). The test was conducted under the same conditions employed for manuscript Fig. 3e, i.e., [KOH] = 1 M and $S = 1 \text{ cm}^2$, except that i_{cell} was 0.90 A for HK-WWE2 as determined by our model to maintain ΔC at 1 M. This adjustment of i_{cell} was necessary since we used a different wastewater effluent than the HK-WWE1 used in Fig. 3e, which resulted in a slight change in membrane permeability. The system operation was continuously monitored over 168 h (7 days) with no interruption and the results for the changes in water flux, voltage and ΔC are shown in Fig. 3f. The water flux decreased by ~12% from an average of 15.6 LHM on the first day to an average of 13.7 LHM by the last day, while the voltage of the system increased by only 4%, starting from 1.75 V and was consistently remained at 1.82 V. Meanwhile, the system sustained a consistent ΔC at around 1 M during the entire 168-h testing period. The 12% reduced water flux after the long-term operation was attributed to fouling from organic matter in the wastewater⁶, because the ratio of C to O elements on the membrane surface (C/O ratio) was reduced by ~13%, and no degradation of the integrity for both the selective and support layers of the FO membrane was observed in SEM (Tescan MAIA3) coupled with Energy Dispersive X-ray (EDX) (Supplementary Fig. 8 and Table 3, respectively).

More importantly, FO membrane fouling has been shown to be largely reversible, unlike irreversible fouling challenges faced in reverse osmosis⁷. The reduced water flux in our system can also be easily recovered by backwashing or chemical washing. Additionally, the measured H₂ gas purity over 99% throughout the entire operation suggests that wastewater matrices did not affect the gas purity even after long-term operation (Supplementary Fig. 9). Therefore, the system performance remained highly stable throughout the 168-h testing period, indicating a successful long-term operation of our integrated system.

These are added to **the newly revised Manuscript:**

Lines 250-265: “A continuous 168-hour operation of the FOWS_{AWE} system was conducted to assess its long-term stability under consistent conditions using a wastewater effluent sample collected from a second wastewater treatment plant in Hong Kong (HK-WWE2), and the result is shown in Fig. 3f. The water flux decreased by ~12% from 15.6 LMH to 13.7 LMH, whereas the voltage increased by 4% during the initial 24 hours, starting from 1.75 V and maintaining at 1.82 V. In addition, the system sustained a ΔC consistently at around 1 M during the entire 168-h testing period. The 12% reduced water flux after the long-term operation was attributed to fouling from organic matter in the wastewater⁴², because the ratio of C to O elements on the membrane surface was reduced by ~13%, and no degradation of integrity for both the selective and support layers of the FO membrane was observed in SEM coupled with Energy Dispersive X-ray (Supplementary Table 3 and Fig. 8, respectively). FO membrane fouling was shown to be largely reversible, unlike the often irreversible fouling challenges faced in reverse osmosis⁴³, which means that the reduced water flux in our system can be easily recovered by backwashing or chemical washing. Notably, the H₂ gas purity remained stably high at over 99% throughout the entire long-term operation (Supplementary Fig. 9). These results demonstrated the operational stability of the FOWS_{AWE} system throughout the 168-h testing period.”

The figure showing the water flux, voltage, and ΔC during the 168-h continuous operation of the integrated FOWS_{AWE} system have been added to **the newly revised Manuscript Fig. 3f.**

The SEM, H₂ purity, and EDX characterization for the FO membrane before and after 168-h operation are added to **the revised Supplementary Fig. 8, Fig. 9, and Table 3**, respectively.

References:

6. Melián-Martel, N., Sadhwani, J. J., Malamis, S. & Ochsenkühn-Petropoulou, M. Structural and chemical characterization of long-term reverse osmosis membrane fouling in a full scale desalination plant. *Desalination* **305**, 44–53 (2012).
7. Lee, S., Boo, C., Elimelech, M. & Hong, S. Comparison of fouling behavior in forward osmosis (FO) and reverse osmosis (RO). *J. Memb. Sci.* **365**, 34–39 (2010).

Figure 3f. Water flux, voltage, and ΔC during the 168-h continuous operation of the integrated FOWS_{AWF} system. (Conditions: $[\text{KOH}] = 1 \text{ M}$, $S = 1 \text{ cm}^2$, and $i_{\text{cell}} = 0.90 \text{ A}$ using the HK-WWE2).

Supplementary Figure 8. SEM images showing **a–b.** the selective layer of a pristine TFC-FO membrane and the TFC-FO membrane after 168 h of operation, respectively; **c–d.** the support layer

of a pristine TFC-FO membrane and the TFC-FO membrane after 168 h of operation, respectively. (Conditions: $[\text{KOH}] = 1 \text{ M}$, $S = 1 \text{ cm}^2$, $i_{\text{cell}} = 0.90 \text{ A}$ for using HK-WWE2).

Supplementary Figure 9. H₂ purity measurements during 168 hours of continuous FOWS_{AWE} system operation. (Conditions: $[\text{KOH}] = 1 \text{ M}$, $S = 1 \text{ cm}^2$, $i_{\text{cell}} = 0.90 \text{ A}$ for using HK-WWE2).

Supplementary Table 3. EDX showing the C and O element and their ratio for the selective layer of a pristine TFC-FO membrane and the TFC-FO membrane after 168 h of operation.

TFC-FO membrane	Element	Mass Norm (%)	Atom (%)	Mass Norm (%)
Pristine	C	82.08	85.92	3.80
	O	17.92	14.08	1.00
	C/O Ratio	4.58	6.10	/
After 168-h use	C	80.00	84.18	3.59
	O	20.00	15.82	0.92
	C/O Ratio	4.00	5.32	/

Responses to comments of Reviewer #2:

Developing membrane-based water electrolysis system which can be used for different water source is important for using renewable energy to produce green hydrogen. This manuscript submitted by Ling et al. reports a modularized forward osmosis-alkaline water electrolysis system to allow for using wastewater to generate high pure hydrogen. The authors suggest that the integrated water electrolysis system enables to use 1 M potassium hydroxide as a draw solution and electrolyte to split water and produce hydrogen. Moreover, the water-hydrogen balance model was established to balance the production and consumption of water based on the reasonable calculations. The integrated system exhibited an excellent H₂ production rate of 448 Nm³ day⁻¹ 99 per m² of membrane area with energy consumption as low as 3.96 kWh Nm⁻³. This work provides a promising strategy for energy-efficient hydrogen generation in areas where water resources are scarce and seawater is unavailable. Therefore, I recommend to publish this manuscript after minor revisions. Below is a list of specific comments and questions the authors should address before publication.

Comment 1:

In this manuscript, the permeate and rejection tests were performed in 1 M KOH mixture consisting of NaAc, NH₃/NH₄⁺ and chloride. In fact, some impurities including phosphorous, trace metal ions and organics also containing in wastewater, which may have an effect on the extraction of water from wastewater. Therefore, the experiments need to be performed to evaluate the impact of above impurities on the TFC-FO membranes and the water flux.

Reply: We appreciate the reviewer for this valuable suggestion. To evaluate other impurities than the original NaAc, NH₃/NH₄⁺ and chloride, we collected real wastewater samples containing a wider range of impurities from different regions as detailed in Supplementary Table 4. These wastewater samples show phosphate reaching 0.4 mg/L, various other inorganic impurities such as bromide and sulfate at 0.01–1.40 mg/L, calcium, magnesium as high as 110.78 mg/L, and total organic carbon (TOC) up to 45 mg/L. Additional FO tests were conducted using these wastewater samples as the feed solution under the conditions of [KOH] at 1 M and *S* at 1 cm². Supplementary Fig. 11a shows that the water flux crossing the FO membrane remains constant during a 5-h testing period in the presence of the inorganic and organic impurities in the feed wastewater samples. After the FO operation, the impurities in the feed samples of GD-WWE and NC-WWE were measured to obtain their FO rejection rates as shown in Supplementary Fig. 5, in which the FO membrane maintained rejection rates over 90% for all those impurities. Additionally, the FO membrane used

for GD-WWE was characterized after the 5-hour test by a high-resolution SEM (Tescan MAIA3) as shown in Supplementary Fig. 12, which indicated no change in its selective layer surface compared to a pristine FO membrane. These results indicate that the FO process only has a minor impact from diverse impurities found in real-world wastewater or wastewater effluent during water extraction.

These are added to **the newly revised Manuscript:**

Lines 172–174: “The rejection of additional common impurities found in wastewater and varied pH levels were also examined, with rejection rates exceeding 90% (Supplementary Figs. 5 and 6).”

Lines 265–269: “In addition, as the FOWS_{AWE} system relies on the water flux to generate H₂, we tested more wastewater samples from the southern and northeastern regions of China, with their composition and concentrations detailed in Supplementary Table 4. The results show that our system exhibited a constant water flux, voltage, and ΔC for various impurities and pH values from different wastewater samples (Supplementary Figs. 10–12).”

The table showing the quality of the collected wastewater samples from different regions is added to **the revised Supplementary Table 4**. The results of rejection rates, water flux, and SEM images of the FO membranes for feeding wastewater samples in the presence of a wide range of impurities are added to **the revised Supplementary Figs. 5, 11a, and 12**, respectively.

Supplementary Table 4. Quality of the collected wastewater effluents and raw wastewater.

Parameter	HK-WWE1 ⁽ⁱ⁾	HK-WWE2 ⁽ⁱⁱ⁾	GD-WWE ⁽ⁱⁱⁱ⁾	NC-WWE ^(iv)
Conductivity (mS/cm)	N.M.*	0.89	4.87	1.44
TOC (mg/L as C)	5.23	18.21	20.53	44.36
pH	7.2	7.85	7.42	7.29
Color (PtCo)	N.M.*	46.00	86.00	37
Ammonia (mg/L as N)	1.09	0.48	0.59	0.12
Chloride (mg/L)	150.2	1.27	9.42	2.56
Nitrite (mg/L)	0.37	0.03	0.01	0.14
Bromide (mg/L)	N.D.*	N.D.*	0.03	0.02
Nitrate (mg/L)	4.43	0.05	0.37	0.28
Sulfate (mg/L)	N.M.*	0.23	1.40	0.68

Phosphate (mg/L)	N.M.*	0.19	0.38	0.22
Calcium (mg/L)	N.M.*	29.24	74.25	101.14
Magnesium (mg/L)	N.M.*	4.35	110.78	28.87
Zinc (mg/L)	N.M.*	0.07	0.06	N.D.
Copper (mg/L)	N.M.*	0.16	0.16	0.06
Iron (mg/L)	N.M.*	0.10	0.09	0.05
Aluminum (mg/L)	N.M.*	0.18	0.15	0.07

*N.D.: not detected

*N.M.: not measured

*(i) HK-WWE1: wastewater effluent collected from Hong Kong; (ii) HK-WWE2: wastewater effluent collected from Hong Kong; (iii) GD-WWE: wastewater effluent collected from Guangdong province; and (iv) NC-WWE: wastewater effluent collected from Northeast China.

Supplementary Fig. 5. The rejection rates of the TFC-FO membrane for feeding different wastewater samples in presence of a wide range of impurities. (Conditions: $[\text{KOH}] = 1 \text{ M}$, $S = 1 \text{ cm}^2$, operation duration = 5 h, $i_{\text{cell}} = 1.05 \text{ A}$ and 1.03 A for GD-WWE and NC-WWE, respectively, the species and content of impurity are detailed in Supplementary Table 4).

Supplementary Fig. 11a. The water flux of the integrated FOWS_{AWE} system when using different wastewater sources. (Conditions: [KOH] = 1 M, $S = 1 \text{ cm}^2$, operation duration = 5 h, $i_{\text{cell}} = 1.08 \text{ A}$, 1.05 A, and 1.03 A for HK-WWE2, GD-WWE, and NC-WWE, respectively).

Supplementary Fig. 12. SEM images showing the selective layers of **a.** the pristine TFC-FO membrane, and **b.** the TFC-FO membrane after 5-h operation using GD-WWE.

Comment 2:

The pH of wastewater also has an effect on TFC-FO membranes, the water flux, and the hydrogen production. Please supply the permeate and rejection tests in wastewater with different pH.

Reply: We appreciate this comment regarding the effect of varying pH presented in wastewater, which can impact on membrane transport properties and solute rejection in FO⁷. We conducted

additional FO tests using GD-WWE (water quality is shown in Supplementary Table 4) while varying the pH level to evaluate the pH effects on the water flux through the FO membrane and the rejection of 14 impurities, chloride, nitrite, bromide, nitrate, sulfate, phosphate, zinc, copper, iron, aluminum, calcium, magnesium, ammonia, and TOC. Three conditions were evaluated - pH 6.5, 7.5, and 8.5 - which encompass the typical range observed in wastewater^{8,9}. The tests were conducted for conditions of [KOH] at 1 M and S at 1 cm². The results demonstrate that the water flux remained stable at ~18 LHM, and is thus not affected by varying the pH of the feed wastewater effluent, as shown in Supplementary Fig. 10. This finding is consistent with previously reported pH effects on TFC membranes^{10,11}. Moreover, the FO membrane maintained high rejection rates of the above impurities over the examined pH range (Supplementary Fig. 6).

These are added to **the newly revised Manuscript:**

Lines 265–269: “In addition, as the FOWS_{AWE} system relies on the water flux to generate H₂, we tested more wastewater samples from the southern and northeastern regions of China, with their composition and concentrations detailed in Supplementary Table 4. The results show that our system exhibited a constant water flux, voltage, and ΔC for various impurities and pH values from different wastewater samples (Supplementary Figs. 10–12).”

These results showing the effect of wastewater effluent pH on the impurity rejection and water flux of FO are added to **the revised Supplementary Fig. 6 and 10**, respectively.

References:

8. Odjadjare, E. E. O. & Okoh, A. I. Physicochemical quality of an urban municipal wastewater effluent and its impact on the receiving environment. *Environ. Monit. Assess.* **170**, 383–394 (2010).
9. Li, B. & Zhang, T. pH significantly affects removal of trace antibiotics in chlorination of municipal wastewater. *Water Res.* **46**, 3703–3713 (2012).
10. Van Wagner, E. M., Sagle, A. C., Sharma, M. M. & Freeman, B. D. Effect of crossflow testing conditions, including feed pH and continuous feed filtration, on commercial reverse osmosis membrane performance. *J. Memb. Sci.* **345**, 97–109 (2009).
11. Arena, J. T., Chwatko, M., Robillard, H. A. & McCutcheon, J. R. pH Sensitivity of Ion

Exchange through a Thin Film Composite Membrane in Forward Osmosis. *Environ. Sci. Technol. Lett.* **2**, 177–182 (2015).

Supplementary Figure 1. The effect of initial wastewater effluent pH on impurity rejection rates in the FO process. Measurements were carried out using GD-WWE with the initial pH adjusted to 6.5, 7.5, and 8.5. (Conditions: $[\text{KOH}] = 1 \text{ M}$, $S = 1 \text{ cm}^2$, $i_{\text{cell}} = 1.05 \text{ A}$).

Supplementary Figure 2. The effect of initial wastewater effluent pH on the water flux. Measurements over time were carried out using GD-WWE with the initial pH adjusted to 6.5, 7.5, and 8.5. (Conditions: $[\text{KOH}] = 1 \text{ M}$, $S = 1 \text{ cm}^2$, $i_{\text{cell}} = 1.05 \text{ A}$).

Long-term durability is an important indicator to evaluate the potential for practical application of the water electrolysis system. The stability test for the longer time is necessary in this work.

Reply: We appreciate the reviewer's comment and acknowledge the importance of long-term operational studies. In response, we conducted a 168-h continuous test using our integrated system fed with a real wastewater effluent obtained from a second Hong Kong wastewater treatment plant (HK-WWE2, its water qualities shown in Supplementary Table 4). The test was conducted under the same conditions employed for manuscript Fig. 3e, i.e., $[\text{KOH}] = 1 \text{ M}$ and $S = 1 \text{ cm}^2$, except that i_{cell} was 0.90 A for HK-WWE2 as determined by our model to maintain ΔC at 1 M. This adjustment of i_{cell} was necessary since we used a different wastewater effluent than the HK-WWE1 used in Fig. 3e, which resulted in a slight change in membrane permeability. The system operation was continuously monitored over 168 h (7 days) with no interruption and the results for the changes in water flux, voltage and ΔC are shown in Fig. 3f. The water flux decreased by $\sim 12\%$ from an average of 15.6 LHM on the first day to an average of 13.7 LHM by the last day, while the voltage of the system increased by only 4%, starting from 1.75 V and was consistently remained at 1.82 V. Meanwhile, the system sustained a consistent ΔC at around 1 M during the entire 168-h testing period. The 12% reduced water flux after the long-term operation was attributed to fouling from organic matter in the wastewater⁶, because the ratio of C to O elements on the membrane surface (C/O ratio) was reduced by $\sim 13\%$, and no degradation of the integrity for both the selective and support layers of the FO membrane was observed in SEM (Tescan MAIA3) coupled with Energy Dispersive X-ray (EDX) (Supplementary Fig. 8 and Table 3, respectively).

More importantly, FO membrane fouling has been shown to be largely reversible, unlike irreversible fouling challenges faced in reverse osmosis⁷. The reduced water flux in our system can also be easily recovered by backwashing or chemical washing. Additionally, the measured H₂ gas purity over 99% throughout the entire operation suggests that wastewater matrices did not affect the gas purity even after long-term operation (Supplementary Fig. 9). Therefore, the system performance remained highly stable throughout the 168-h testing period, indicating a successful long-term operation of our integrated system.

These are added to **the newly revised Manuscript:**

Lines 250-265: “A continuous 168-hour operation of the FOWS_{AW}E system was conducted to assess its long-term stability under consistent conditions using a wastewater effluent sample collected from a second wastewater treatment plant in Hong Kong (HK-WWE2), and the result is

shown in Fig. 3f. The water flux decreased by ~12% from 15.6 LMH to 13.7 LMH, whereas the voltage increased by 4% during the initial 24 hours, starting from 1.75 V and maintaining at 1.82 V. In addition, the system sustained a ΔC consistently at around 1 M during the entire 168-h testing period. The 12% reduced water flux after the long-term operation was attributed to fouling from organic matter in the wastewater⁴², because the ratio of C to O elements on the membrane surface was reduced by ~13%, and no degradation of integrity for both the selective and support layers of the FO membrane was observed in SEM coupled with Energy Dispersive X-ray (Supplementary Table 3 and Fig. 8, respectively). FO membrane fouling was shown to be largely reversible, unlike the often irreversible fouling challenges faced in reverse osmosis⁴³, which means that the reduced water flux in our system can be easily recovered by backwashing or chemical washing. Notably, the H₂ gas purity remained stably high at over 99% throughout the entire long-term operation (Supplementary Fig. 9). These results demonstrated the operational stability of the FOWS_{AWE} system throughout the 168-h testing period.”

The figure showing the water flux, voltage, and ΔC during the 168-h continuous operation of the integrated FOWS_{AWE} system have been added to **the newly revised Manuscript Fig. 3f**.

The SEM, H₂ purity, and EDX characterization for the FO membrane before and after 168-h operation are added to **the revised Supplementary Fig. 8, Fig. 9, and Table 3**, respectively.

References:

6. Melián-Martel, N., Sadhwani, J. J., Malamis, S. & Ochsenkühn-Petropoulou, M. Structural and chemical characterization of long-term reverse osmosis membrane fouling in a full scale desalination plant. *Desalination* **305**, 44–53 (2012).
7. Lee, S., Boo, C., Elimelech, M. & Hong, S. Comparison of fouling behavior in forward osmosis (FO) and reverse osmosis (RO). *J. Memb. Sci.* **365**, 34–39 (2010).

Figure 3f. Water flux, voltage, and ΔC during the 168-h continuous operation of the integrated FO_{SAWE} system. (Conditions: $[\text{KOH}] = 1 \text{ M}$, $S = 1 \text{ cm}^2$, and $i_{\text{cell}} = 0.90 \text{ A}$ using the HK-WWE2)

Supplementary Figure 3. SEM images showing **a–b.** the selective layer of a pristine TFC-FO membrane and the TFC-FO membrane after 168 h of operation, respectively; **c–d.** the support layer of a pristine TFC-FO membrane and the TFC-FO membrane after 168 h of operation, respectively. (Conditions: $[\text{KOH}] = 1 \text{ M}$, $S = 1 \text{ cm}^2$, $i_{\text{cell}} = 0.90 \text{ A}$ for using HK-WWE2).

Supplementary Figure 9. H₂ purity measurements during 168 hours of continuous FOWSAWE system operation. (Conditions: [KOH] = 1 M, $S = 1 \text{ cm}^2$, $i_{\text{cell}} = 0.90 \text{ A}$ for using HK-WWE2).

Supplementary Table 3. EDX showing the C and O element and their ratio for the selective layer of a pristine TFC-FO membrane and the TFC-FO membrane after 168 h of operation.

TFC-FO membrane	Element	Mass Norm (%)	Atom (%)	Mass Norm (%)
Pristine	C	82.08	85.92	3.80
	O	17.92	14.08	1.00
	C/O Ratio	4.58	6.10	/
After 168-h use	C	80.00	84.18	3.59
	O	20.00	15.82	0.92
	C/O Ratio	4.00	5.32	/

Comment 4:

The species and content of impurities in wastewater vary in different regions. In order to make this integrated water electrolysis system more convincing, the H₂ generation rate tests need to be performed in the wastewater effluent sample which are collected from the different regions.

Reply: We appreciate for this valuable suggestion. Indeed, the composition and concentration of impurities in wastewater vary among different regions. To evaluate our integrated system more comprehensively under various real-world wastewater conditions, we collected additional wastewater samples from the southern and the northeastern regions of China, rather than only using HK-WWE1 in the former manuscript. These additional wastewater samples include a sample

collected in a second wastewater plant in Hong Kong (HK-WWE2), a wastewater effluent sample collected in Guangdong province (GD-WWE), and a wastewater effluent sample collected in northeast China (NC-WWE). Their compositions are detailed on Supplementary Table 4. The performance of our system using these wastewater samples was conducted under the same conditions employed for HK-WWE1 (Fig. 3e), except i_{cell} was 1.08 A for HK-WWE2, 1.05 A for GD-WWE, and 1.03 A for NC-WWE for maintaining ΔC of 1 M. The tests were conducted for a 5-h period. The results demonstrated our system maintained stable water flux, voltage, and ΔC across the range of wastewater compositions tested, as shown in Supplementary Figure 11. In addition, the H_2 production rate values were 59.2, 57.6, and 56.5 L day⁻¹ for HK-WWE2, GD-WWE, and NC-WWE, respectively, being comparable to results for the HK-WWE1. These results provide strong evidence that the integrated FOWS_{AWE} system can effectively produce hydrogen from wastewater regardless of the local wastewater quality characteristics. Collecting wastewater samples from various regions as suggested by the reviewer, enables a more comprehensive performance assessment that better reflects the diverse real-world feed conditions. We believe these additional tests significantly strengthen the applicability and practical viability of our proposed system not only for sustainable hydrogen production but also for wastewater treatment and resource recovery applications in different regions.

These are added to **the newly revised Manuscript:**

Lines 265–272: “In addition, as the FOWS_{AWE} system relies on the water flux to generate H_2 , we tested more wastewater samples from the southern and northeastern regions of China, with their composition and concentrations detailed in Supplementary Table 4. The results show that our system exhibited a constant water flux, voltage, and ΔC for various impurities and pH values from different wastewater samples (Supplementary Figs. 10–12). Despite the varied $K_w \times I$ caused by different impurities, the resulted H_2 production rates were consistent and remained within a range of 5% (Supplementary Fig. 11d). These results validate the stability and adaptability of our system for producing H_2 under varying wastewater conditions.”

Former text in Lines 211–213: “In this study, Φ for feeding deionised water and wastewater effluent with an FO-AWE integrated system were calculated as 1.55 and 0.95 A mol⁻¹ cm⁻², respectively.” was revised as:

Lines 221-224: “In this study, Φ values for feeding deionised water and the wastewater effluent collected from a wastewater treatment plant in Hong Kong (HK-WWE1) to the FOWS_{AWE} system were calculated as 1.55 and 0.95 A mol⁻¹ cm⁻², respectively.”

Former text in Lines 216–217: “Therefore, to maintain a ΔC at 1 M, J_i was calculated as 1.55 and 0.95 A cm⁻² for deionised water and wastewater effluent, respectively.” was revised as:

Lines 227-228: “Therefore, to maintain ΔC at 1 M, J_i was calculated as 1.55 and 0.95 A cm⁻² for deionised water and HK-WWE1, respectively²¹.”

Former text in Lines 218–220: “The two setups of FO and AWE were then experimentally integrated using a 1 cm² FO membrane and applying i_{cell} of AWE at 1.55 and 0.95 A for deionised water and real wastewater effluent according to their J_i values, respectively.” was revised as:

Lines 229-231: “The two setups of FO and AWE were then experimentally integrated using a 1 cm² FO membrane and applying an i_{cell} of AWE of 1.55 and 0.95 A for deionised water and HK-WWE1 according to their J_i values, respectively.”

The comparison in water flux, voltage, ΔC , and H₂ production rates of the integrated system when using wastewater samples from different regions is added to **the revised Supplementary Fig. 11.**

Supplementary Figure 11. **a.** Water flux; **b.** voltage, **c.** ΔC between feed and draw solution, and **d.** H_2 flux of the integrated FOWSAWE system when using different wastewater sources. (Conditions: $[KOH] = 1 \text{ M}$, $S = 1 \text{ cm}^2$, $i_{\text{cell}} = 1.08 \text{ A}$, 1.05 A , and 1.03 A for HK-WWE2, GD-WWE, and NC-WWE, respectively).

Comment 5:

There are some vague description and typo errors in this article. Thus, for a better readability, please check throughout the whole article text and correct all the errors.

Reply: We appreciate the feedback on improving the clarity and readability of our manuscript. We conducted a thorough revision of the entire text to address any inconsistencies or errors for maintaining high standards of accuracy and clarity. The changes are marked in blue in **the newly revised Manuscript**.

REVIEWERS' COMMENTS

Reviewer #1 (Remarks to the Author):

The comments from the initial review have been addressed. The adaptation of FOWS nomenclature is acceptable. There was no need to clutter the literature with multiple names of the same concept. The subscript indicates how FOWS was performed ... thus it keeps the FOWS nomenclature and at the same time distinguishes the contribution by the authors. So, well done.

The work is very technically sound and carefully done. The paper is therefore ready for publication.

Reviewer #2 (Remarks to the Author):

The authors have answered all the questions and this manuscript can now be accepted in the present form